# Clinical Characteristics of Infants with Symptomatic Congenital and Postnatal Cytomegalovirus Infection—An 11-Year Multicenter Cohort Study in Taiwan

**DOI:** 10.3390/children11010017

**Published:** 2023-12-22

**Authors:** Yu-Ning Chen, Kai-Hsiang Hsu, Chung-Guei Huang, Ming-Chou Chiang, Shih-Ming Chu, Chyi-Liang Chen, Jen-Fu Hsu, Ho-Yen Chueh

**Affiliations:** 1Division of Neonatology, Department of Pediatrics, Chang Gung Memorial Hospital, Taoyuan 33382, Taiwan; eunice80221@cgmh.org.tw (Y.-N.C.); khsu@cgmh.org.tw (K.-H.H.); cmc123@cgmh.org.tw (M.-C.C.); kz6479@cgmh.org.tw (S.-M.C.); hsujanfu@cgmh.org.tw (J.-F.H.); 2College of Medicine, Chang Gung University, Taoyuan 33302, Taiwan; 3Graduate Institute of Clinical Medical Science, Chang Gung University, Taoyuan 33302, Taiwan; 4Department of Laboratory Medicine, Chang Gung Memorial Hospital, Taoyuan 33382, Taiwan; joyce@cgmh.org.tw; 5Department of Medical Biotechnology and Laboratory Science, College of Medicine, Chang Gung University, Taoyuan 33302, Taiwan; 6Molecular Infectious Disease Research Center, Chang Gung Memorial Hospital, Taoyuan 33382, Taiwan; dinoschen@cgmh.org.tw; 7Department of Obstetrics and Gynaecology, Chang Gung Memorial Hospital, Chang Gung University, Taoyuan 33302, Taiwan

**Keywords:** cytomegalovirus, congenital, postnatal, infants

## Abstract

(1) Background: Cytomegalovirus (CMV) infection is a prevalent viral disease among infants. The prevalence typically ranges from 0.2% to 2.4% among all newborns. There are limited data regarding the demographic characteristics of infants with symptomatic CMV infections. (2) Methods: In this retrospective cohort study using the Chang Gung Memorial Hospital multicenter database, infants with CMV infection determined by a positive urine culture, positive blood polymerase chain reaction assay or positive immunoglobulin M result for CMV from 2011 through 2021 were included. Clinical characteristics at initial diagnosis, management and outcomes were investigated. Congenital CMV (cCMV) infection is diagnosed within three weeks after birth; postnatal CMV (pCMV) is diagnosed when CMV is detected after the first 3 weeks of life. (3) Results: Among the 505 CMV-infected infants identified, 272 were included in the analysis. According to the age at initial presentation, 21 infants had cCMV infection and 251 had pCMV infection. Higher incidences of prematurity and being small for gestational age and a lower Z score for weight at diagnosis were observed in the cCMV group. While thrombocytopenia (61.9%) was the leading presentation in the cCMV group, hepatitis (59.8%) and prolonged jaundice (21.9%) were more common in the pCMV group. (4) Conclusions: Utilizing an 11-year multicenter database, we demonstrated the characteristics of infants with CMV infection in Taiwan and highlighted the demographic disparities and differing symptoms between the cCMV and pCMV groups. These findings emphasize the necessity for future research to refine screening policies, explore treatment options, and establish follow-up protocols for affected infants.

## 1. Introduction

Human cytomegalovirus (CMV) is classified within the Herpesviridae family, identified as a double-stranded DNA virus. This infection is widespread among infants and constitutes a considerable source of morbidity, occasionally leading to mortality among neonates. In earlier research, it was found that CMV prevalence can be influenced by factors such as age, culture, and socioeconomic status. Even in developed countries like the United States or the United Kingdom, CMV infections happen throughout childhood and adolescence, affecting around 60 to 80 percent of the population by adulthood [1]. CMV infection can be transmitted through different routes: transplacentally (known as congenital CMV or cCMV, which is diagnosed within the first three weeks of life), through cervical or vaginal secretions, or via breast milk, mucosal secretions or blood transfusion (termed postnatal CMV or pCMV, which is diagnosed after the first three weeks of life but within the first year) [2,3,4].

The occurrence of cCMV infection varies depending on the overall CMV prevalence in the population and maternal socioeconomic status. The prevalence typically ranges from 0.2% to 2.4% among all newborns [5]. The classification of cCMV into symptomatic or asymptomatic categories lacks a clear-cut definition in the existing literature. Broadly, moderately to severely symptomatic cCMV refers to infants with infections showing multiple manifestations or involvement of the central nervous system (CNS). Mildly symptomatic cCMV entails infants with one or two isolated manifestations that are mild and short-lived. Asymptomatic cCMV with isolated sensorineural hearing loss (SNHL) characterizes infants displaying no evident clinical symptoms beyond hearing loss. Meanwhile, asymptomatic cCMV describes infants without apparent abnormalities at birth and who exhibit normal hearing [6]. Among newborns with cCMV infection, only approximately 10% to 15% display noticeable symptoms [7]. Among these symptomatic infants, the mortality rate can be up to 30%, and they are at risk of permanent sequelae, with sensorineural hearing loss (SNHL) being the most common, followed by cognitive impairment, motor deficits, and chorioretinitis, which occur in approximately 40% to 58% of cases. Even among asymptomatic cCMV infants, approximately 12% to 13.5% may still be complicated with neurodevelopmental or hearing disabilities [8,9,10].

pCMV infection has been observed with a median incidence of approximately 20%, although this rate varies widely, ranging from 6% to 59% [10]. The occurrence of symptomatic pCMV infection also shows significant variability. While most infants with pCMV infection may be asymptomatic, severe cases (e.g., sepsis-like syndrome) have been reported. The severity of such cases appears to be directly related to the infant’s maturity and health status; preterm infants, notably those classified as very low birth weight (VLBW) with a birth weight below 1500 g, face increased susceptibility to severe CMV disease. The infection can manifest as early as three weeks after birth or as late as three to six months of age. Early transmission of the virus stands as a risk factor contributing to symptomatic disease in these infants. [2,11,12,13]. Distinguishing between congenital and postnatal CMV infections poses the primary challenge in diagnosing CMV during the perinatal period. While any newborn displaying signs or symptoms suggestive of CMV infection should undergo testing, in Taiwan, the routine practice of neonatal CMV screening is absent, and there is a lack of established guidelines for conducting CMV testing in infants, resulting in a scarcity of comprehensive data on the demographic profiles of infants affected by symptomatic cCMV or pCMV infections, including information on their symptoms and outcomes. Therefore, the primary objective of this study was to explore and investigate cases of symptomatic CMV infection in infants across multiple medical centers in Taiwan.

## 2. Materials and Methods

### 2.1. Study Design

This research utilized a retrospective cohort extracted from the Chang Gung Memorial Hospital central laboratory database, encompassing data from various centers, namely Keelung, Linkou, Taoyuan, Chiayi, and Yunlin branches, covering the period from January 2011 to December 2021. Infants diagnosed with CMV infection before reaching one year of age were included in the study. Their demographic details, clinical characteristics at the time of initial diagnosis, treatment approaches, and outcomes were thoroughly examined. The demographic and clinical characteristics of infants were documented by the clinicians who were primarily responsible for the infants’ care and were recorded in the medical records during the period of the infants’ care and treatment. Exclusion criteria involved infants lacking CMV-related symptoms and those with incomplete clinical records. The study received approval from the Institutional Review Board and Human Research Ethics Committee (IRB Number: 202001461B0), with an authorized waiver for obtaining informed consent due to the collection of anonymized data.

### 2.2. Definitions of CMV Infection

CMV infection was defined as either having a documented positive urine culture, a positive blood polymerase chain reaction (PCR) assay or a positive immunoglobulin M (IgM) result for CMV. Despite the lower sensitivity of CMV-IgM in detecting CMV infection, it is noteworthy that the positive predictive value for CMV-IgM reached as high as 96.4% [14]. This level of accuracy remains valuable, especially in resource-limited settings where CMV-PCR is either unavailable or deemed too costly. Infants had to display any of the following findings suggestive of CMV infection: intrauterine growth restriction (IUGR, defined as estimated fetal weight less than 10th percentile [15,16]), small for gestational age (SGA, birthweight < 10th percentile according to the individual’s gestational age), microcephaly (head circumference ≤ 2 standard deviations for gestational age), prolonged jaundice (jaundice > 14 days of life), petechiae or purpura, or hepatosplenomegaly. Perinatal CMV exposure was specifically identified as a confirmed active CMV infection in the mother. Neurological abnormalities associated with CMV infection encompass several conditions: microcephaly, defined by a head circumference more than 2 standard deviations below the mean for age and sex or less than the 3rd percentile for age and sex [17,18]; encephalitis; sensorineural hearing loss, characterized by a failure in Automated Auditory Brainstem Responses (AABR) or Auditory Brain Stem Response (BSR) tests; seizures; and developmental delay, denoted by a delay or failure to reach milestones in one or more developmental domains (such as communication, motor skills, cognition, social-emotional skills, or adaptive skills) in line with a child’s expected developmental progression for their age [19]. Additionally, laboratory findings, such as neutropenia (less than 1500 cells/mm^3^), thrombocytopenia (platelet count less than 100,000/mm^3^) and elevated levels of serum transaminases, were also considered laboratory indicators of CMV infection [2,10,20,21].

### 2.3. Statistics

Statistical analyses were performed using IBM SPSS Statistics, version 21.0 (IBM, Chicago, IL, USA). For between-group comparisons, continuous data were analyzed using the independent t test, and categorical data were analyzed using the chi-square test or Fisher’s exact test, where appropriate. Statistical significance was defined as *p* < 0.05.

## 3. Results

In our multicenter database from 2011 to 2021, 505 CMV-infected infants were identified; of these, 232 were excluded because of repeated or incomplete data, or because they were asymptomatic at the time the laboratory tests were performed. For the two asymptomatic infants, one underwent self-paid CMV screening and the other was incidentally screened before stem cell transplantation. Therefore, a total of 272 infants were diagnosed with symptomatic CMV infection. According to the age at initial presentation, 21 (7.7%) infants had cCMV infection and 251 had pCMV infection. Furthermore, 140 pCMV infants were diagnosed at <90 days old (young infants), and the remaining 111 were diagnosed after 90 days of age. The enrollment flow diagram is illustrated in Figure 1, and the annual number of cases of CMV infection in infants is displayed in Figure 2.

The demographic characteristics of the cCMV and pCMV groups are presented in Table 1. The majority of cCMV-infected infants were preterm infants (12/21, 57.1%). In addition, more than one-quarter of the cCMV-infected infants experienced IUGR (6/21, 28.6%), and 42.9% (9/21) were SGA. In contrast, only 26.7% (67/251) of the pCMV-infected infants were born prematurely, 5.2% (13/251) had a history of IUGR, and 5.6% (14/251) were SGA (all between-group comparisons, *p* < 0.05). In addition, cCMV-infected infants also exhibited significantly lower body weight z scores than the pCMV-infected infants (−2.16 ± 2.02 versus −0.7 ± 1.40, *p* < 0.001).

The clinical and laboratory features of the cCMV and pCMV groups at diagnosis are summarized in Table 2. While most infants in the cCMV group exhibited thrombocytopenia (13/21, 61.9%) and SGA (9/21, 42.9%), most infants in the pCMV group presented with hepatitis (150/251, 59.8%) and prolonged jaundice (55/251, 21.9%). More specifically, an even greater proportion of infants diagnosed beyond 90 days of age presented with hepatitis (83/111, 74.8%), while fewer infants in this subgroup exhibited prolonged jaundice (6/111, 9.0%) or thrombocytopenia (10/111, 5.4%) (Figure 3).

Regarding treatment for CMV infection, eight (38.0%) infants in the cCMV group had received ganciclovir therapy, while a significantly lower proportion of infants in the pCMV group received antiviral therapy (32/251, 12.7%, *p* = 0.002) (Table 3).

In the cCMV group, the incidence of chorioretinitis, sensorineural hearing loss, and seizures was notably higher than that in the pCMV group (Table 3). Two cCMV-infected infants had intracranial calcification documented during brain sonography and magnetic resonance imaging. We also conducted a comparison of the incidence of chorioretinitis and hearing impairment screening between the two groups. In the cCMV group, 81% (17/21) of the infants had been screened for chorioretinitis and 9.5% (2/17) of them were confirmed to have chorioretinitis. In contrast, only 29.5% (74/251) of the pCMV group underwent screening for chorioretinitis. Regarding hearing tests, 66.7% (14/21) of the cCMV-infected infants had undergone hearing screening, and 23.8% (5/14) of them had a hearing impairment. However, only 20% (50/251) of the pCMV group had undergone hearing screening. None of the infants in the cCMV group died due to CMV infection, but there were 7 CMV-associated mortality cases in the pCMV group after all other potential diagnoses were carefully considered and ruled out.

## 4. Discussion

This study demonstrates the clinical characteristics of infants with CMV infection in Taiwan. The strength of our study lies in the use of a large multicenter database from five medical centers across Taiwan. Additionally, we also demonstrated the difference in demographics and manifestations between infants with cCMV and pCMV infection.

Although we recognized the importance of maternal characteristics and the socioeconomic status of parents, our conclusion, after extensive literature review and consideration of local prevalence data, was that these factors might have a restricted impact within our population due to the high prevalence of CMV in Taiwan. Studies have indicated that CMV seroprevalence among Southeast Asians, including Taiwan, can reach as high as 86% [13]. Moreover, challenges arose in extracting maternal data for a subset of babies who were not delivered at our hospital, further complicating the inclusion of this information. In terms of demographics of infected infants, the ratio of preterm infants, as well as those with IUGR and SGA, was significantly greater in the cCMV group. The outcomes from these cohorts aligned with earlier discoveries suggesting that CMV infection could harm the placenta either through the direct effect of the virus on cells or due to an overwhelming immune response. While there is limited understanding of the immune reactions triggered by CMV in the placenta, it is plausible that an overly active initial inflammatory response or irregularities in the body’s adaptive immune defenses might contribute to tissue damage [22,23]. Our findings highlighted the adverse sequelae of CMV infection on fetal growth [24,25], and a more extended CMV screening program during pregnancy may be necessary [26]. Despite the widespread consensus against prenatal screening for pregnant women due to concerns about causing anxiety, prompting additional tests, and potentially leading to unnecessary pregnancy terminations, there is an inclination toward educating all expectant mothers, regardless of their serostatus, and conducting screenings for all newborns at birth. These approaches are considered more effective in preventing and identifying congenitally CMV-infected children [27,28].

In line with a previous report [29], we found that thrombocytopenia was the most frequently observed symptom in the cCMV group, while hepatitis and prolonged jaundice were more common in the pCMV group. It was important to clinical practitioners that promptly recognizing these leading symptoms could facilitate diagnosis.

According to the most recent guidelines from the Joint Committee on Infant Hearing [30], all infants diagnosed with cCMV infection are considered to be at a high risk of sensorineural hearing loss (SNHL) and should undergo diagnostic testing and receive close audiological follow-up. The risk of developing SNHL is associated with factors such as first-trimester infection (as opposed to infection in the third trimester), symptomatic disease, and intracranial involvement [31,32]. Symptomatic cCMV infection has also been linked to various ophthalmological manifestations, underscoring the importance of annual ophthalmologic evaluations for individuals affected by symptomatic cCMV infection [33]. In contrast, most studies did not consistently show a clear association between pCMV infection and SNHL or several ophthalmologic manifestations [34,35,36] Despite limited evidence, there are indications that pCMV infection might elevate the risk of cognitive and motor developmental deficits as well as hearing impairment [34,37,38,39,40]. In our study, infants in the cCMV group were more likely to undergo audiological follow-up (66.7%), whereas in the pCMV group, only a small portion of infants (20%) underwent audiological follow-up. In terms of ophthalmologic evaluations, it is noteworthy that 81% of the infants in the cCMV group had undergone ophthalmologic assessments. However, only 29.5% of infants in the pCMV group had undergone ophthalmologic evaluations. We recognized that there is a screening bias in our population, and it is crucial to enhance audiologic and ophthalmologic monitoring for CMV-infected infants.

In the context of neurodevelopmental follow-up, there has been a growing emphasis on the wide range of neuroimaging abnormalities observed in infants with cCMV infection. A recent study demonstrated that an MRI severity score serves as a more effective predictor of adverse neurological outcomes than symptoms at birth [41]. Therefore, the clinical guidelines recommend that all cCMV-infected infants initially undergo a cranial ultrasound, and a subsequent MRI is recommended if abnormalities are detected [42,43]. There remains a need for longitudinal follow-up to assess the neurodevelopmental outcomes of infected infants.

Most children will experience late-onset audiologic, neurologic, and developmental sequelae; however, neither the initial physical check at birth nor the hearing screening for newborns is entirely dependable at pinpointing at-risk children. The economic strain stemming from congenital CMV infection is substantial due to the considerable need for ongoing care, specialized therapy, and educational support for many affected children. As a result, there is a pressing need for a dependable, swift, and potentially affordable way to screen newborns for congenital CMV infection. Identifying these infants early on will enable proper monitoring and timely intervention to support their speech and language development [28]. Furthermore, a comprehensive follow-up protocol is crucial for infants affected by CMV.

The primary antiviral treatments recommended as the first line of defense against congenital CMV disease are intravenous (IV) ganciclovir and its oral prodrug, valganciclovir. There is still controversy regarding the use of ganciclovir to treat CMV infection in infancy, and the treatment is currently recommended primarily for infants with severe symptomatic CMV infection that involves the central nervous system, particularly those with an underlying primary immunodeficiency. The commencement of antiviral therapy is recommended immediately upon confirmation through virologic testing. Clinical trials exploring the efficacy of antiviral therapy in congenital CMV disease have shown positive outcomes when treatment begins within the first 30 days after birth [43]. In 2015, Kimberlin et al. conducted a randomized controlled trial comparing a 6-month course of oral valganciclovir (the prodrug of IV ganciclovir) with a 6-week course for infants with symptomatic cCMV infection. Their findings indicated a modest improvement in hearing and neurodevelopmental outcomes at 24 months in the treatment group. Neutropenia caused by drug treatment is most worrisome in the first 6 weeks of therapy, and the risk seems to be lower when using oral valganciclovir as the exclusive treatment [44]. The occurrence of this side effect was notably lower (19%) than what was observed during intravenous ganciclovir treatment in the past (63%) [45]. In our cohort study, it is worth noting that eight (38%) cCMV-infected infants received ganciclovir treatment; each of them commenced treatment within the first 30 days following birth, but only two of them completed the 6-month treatment course. One of these two patients initially received ganciclovir treatment for 3 weeks but had to switch to valganciclovir treatment due to the development of pancytopenia. The second individual completed a comprehensive six-month course of ganciclovir treatment without experiencing any apparent adverse side effects. Additionally, both individuals showed negative findings in subsequent cranial ultrasounds, brain MRIs, ophthalmologic evaluations, and follow-ups in audiology. Among the infants who did not receive 6 months of treatment, two were subsequently diagnosed with SNHL later in life, while another was diagnosed with chorioretinitis. The majority of pCMV infections are asymptomatic, typically not necessitating any form of treatment. Although there are reports of the use of ganciclovir or valganciclovir for pCMV infection with life-threatening symptoms especially in preterm or very low birth-weight (VLBW; birth weight < 1500 g) infants, there is still no large-scale trial focused on treatment for pCMV-infected infants [46,47,48,49]. In our cohort, 12.7% of the pCMV group had received treatment, particularly those diagnosed with pCMV early in their lives, those born prematurely and those presenting with profound thrombocytopenia, hepatitis, or pneumonitis. A comprehensive study is warranted to evaluate the pros and cons of pharmacological treatment in pCMV-infected infants.

Several limitations should be addressed in this cohort study. To begin with, Taiwan lacks a nationwide universal CMV screening program. As a consequence, only infants exhibiting clinical suspicions undergo testing for CMV infection. This approach leads to the potential oversight of asymptomatic cases, resulting in an underestimation of the number of affected infants. Moreover, the absence of comprehensive screening makes it challenging to accurately calculate the overall incidence of neonatal CMV infection or to identify epidemiological differences among various cities in Taiwan. Furthermore, uncertainty persists regarding whether infants testing positive for CMV after surpassing 3 weeks of age had undergone prior negative testing within that time frame to rule out congenital CMV infection. However, it is crucial to highlight that the primary emphasis of this study was to investigate the clinical features apparent at the time of diagnosing CMV infection. We anticipated that this focus would somewhat mitigate the impact of the aforementioned uncertainty. To bridge this gap, there is a need for prospective studies that assess the benefits of routine neonatal CMV screening. These studies would provide essential data for establishing definitive diagnoses, understanding the characteristics of CMV infections, and determining infection rates. Additionally, there is an absence of established guidelines for treatment options and comprehensive surveys addressing CMV-associated sequelae in our country. It is imperative to prioritize efforts aimed at enhancing follow-up procedures for infants diagnosed with CMV infection. These initiatives are crucial for improving long-term outcomes in affected individuals.

## 5. Conclusions

Through the utilization of this multicenter database, we aimed to illustrate the clinical features observed in infants with CMV infection in Taiwan. Our findings revealed a notable increase in the occurrence of premature birth, SGA, and thrombocytopenia among the cCMV group. Moreover, hepatitis and prolonged jaundice emerged as prevalent symptoms in pCMV) infants, particularly in those diagnosed beyond 90 days of age. These findings emphasize the necessity for future investigations to refine screening strategies, evaluate the advantages of routine neonatal CMV screening, delve into treatment possibilities, and establish comprehensive follow-up protocols for affected infants.

## Figures and Tables

**Figure 1 children-11-00017-f001:**
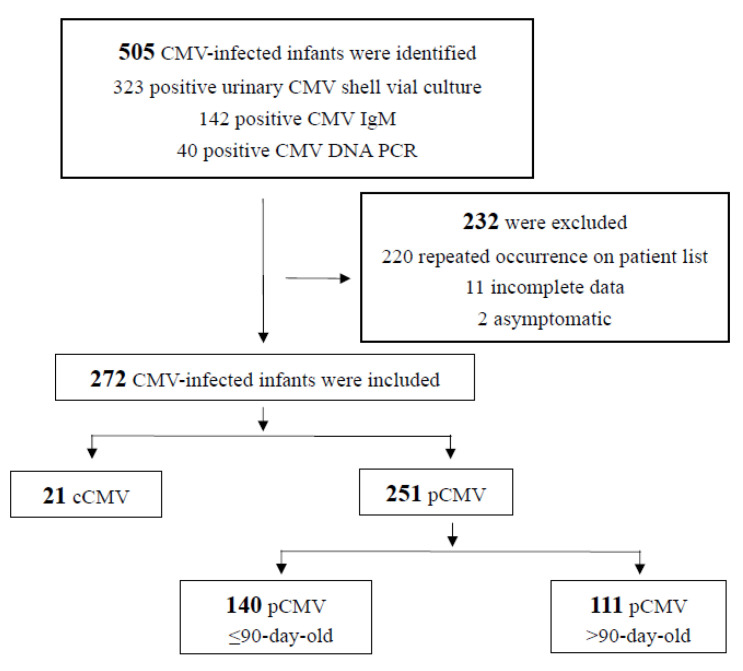
Study cohort flow diagram.

**Figure 2 children-11-00017-f002:**
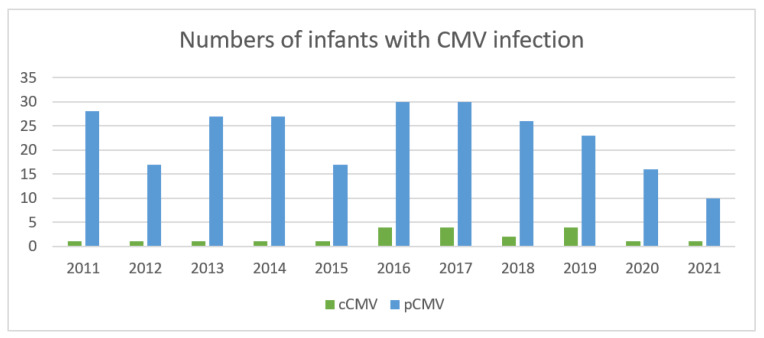
The annual numbers of cases of CMV infection in infants over an 11-year duration.

**Figure 3 children-11-00017-f003:**
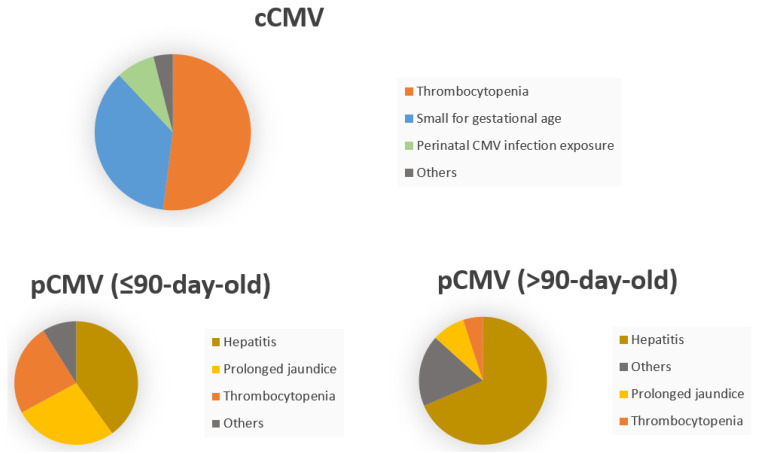
Clinical features at identification of CMV infection Clinical features at diagnosis of CMV infection: The majority of infants in the cCMV group presented with thrombocytopenia and SGA. Meanwhile, most infants in the pCMV group displayed hepatitis and prolonged jaundice. In particular, among infants diagnosed > 90 days of age, a notably higher percentage of infants manifested hepatitis. In contrast, a smaller portion of infants in this subgroup showed prolonged jaundice or thrombocytopenia.

**Table 1 children-11-00017-t001:** Clinical characteristics of the cCMV and pCMV groups.

Characteristics	cCMV Group (*n* = 21)	pCMV Group (*n* = 251)	*p*-Value
Male	10 (47.6%)	147 (58.6%)	0.329
Prematurity	12 (57.1%)	81 (32.3%)	0.021
Vaginal delivery	12 (57.1%)	176 (70.1%)	0.216
Intrauterine growth restriction	6 (28.6%)	13 (5.2%)	<0.001
Small for gestational age	9 (42.9%)	14 (5.6%)	<0.001
Z score for weight at diagnosis	−2.16 (2.02)	−0.7 (1.40)	<0.001

Data are shown as the *n* (%) or mean (SD).

**Table 2 children-11-00017-t002:** Clinical features at the diagnosis of CMV infection of the cCMV and pCMV groups.

	cCMV Group (*n* = 21)	pCMV Group (≤90 Days Old) (*n* = 140)	pCMV Group (>90 Days Old) (*n* = 111)
Thrombocytopenia	13 (61.9%)	40 (28.6%)	6 (5.4%)
Small for gestational age	9 (42.9%)	2 (1.4%)	0
Perinatal CMV infection exposure	2 (9.5%)	4 (2.9%)	0
Hepatitis	0	67 (47.9%)	83 (74.8%)
Prolonged jaundice	0	45 (32,1%)	10 (9.0%)
Positive CMV culture	0	2 (1.4%)	6 (5.4%)
Pneumonitis	0	1 (0.7%)	3 (2.7%)
Neutropenia	0	1 (0.7%)	1 (0.9%)
Failure to thrive	0	1 (0.7%)	0
Colitis	0	1 (0.7%)	0
Petechiae/Purpura	0	0	4 (3.6%)
Myocarditis	0	0	1 (0.9%)
Nephrotic syndrome	0	0	1 (0.9%)
Neurological abnormality			
Microcephaly	1 (4.8%)	0	0
Encephalitis	0	1 (0.7%)	1 (0.9%)
Sensorineural hearing loss	0	1 (0.7%)	0
Seizure	0	0	4 (3.6%)
Developmental delay	0	0	1 (0.9%)
Any neurological abnormality	1 (4.8%)	4 (2.9%)	7 (6.3%)

Data are shown as the *n* (%).

**Table 3 children-11-00017-t003:** Comparison of management, specific screening, and outcomes between the cCMV and pCMV groups.

	cCMV Group (*n* = 21)	pCMV Group (*n* = 251)	*p*-Value
Antiviral treatment	8 (38%)	32 (12.7%)	0.002
Screen for chorioretinitis	17 (81%)	74 (29.5%)	<0.001
Chorioretinitis ^†^	2/17 (11.8%)	1/74 (1.4%)	0.03
Screen of hearing loss	14 (66.7%)	50 (20.0%)	<0.001
Sensorineural hearing loss ^†^	5/14 (35.7%)	4/50 (8.0%)	0.008
Seizure	1 (4.8%)	6 (2.4%)	0.51
Any neurological abnormality	9 (42.9%)	15 (6.0%)	<0.001
CMV-associated mortality	0 (0%)	7 (2.8%)	0.438

Data are shown as the *n* (%). ^†^ Only infants who underwent screening were included.

## Data Availability

The data presented in this study are available on request from the corresponding author. The data are not publicly available due to privacy.

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
