# Peer review of "Clinical Characteristics of Infants with Symptomatic Congenital and Postnatal Cytomegalovirus Infection—An 11-Year Multicenter Cohort Study in Taiwan"

_children, 2023, doi:10.3390/children11010017_

Round 1

Reviewer 1 Report

Comments and Suggestions for Authors

The following comments are related to the review of the manuscript entitled “Clinical Characteristics of Infants with Symptomatic Congenital and Postnatal Cytomegalovirus Infection – A 10-Year Multi-center Cohort Study in Taiwan”. 

The manuscript is appropriately prepared and the authors used the right methodology to achieve their objectives. However, since this is a descriptive study, the level of quality/evidence is low. In any case, as there is always room for improvement in a report, following you will find some comments to be considered. 

Title: In the Methods section, it is stated that the data were collected from January 2011 through December 2021. That is an 11-year period as seen, for instance, in Figure 2. Please review the title and the text. 

In the Results chapter, please avoid to repeat information: do not duplicate the same results in the text and in the tables. For example, the paragraph in lines 90-93 includes the same information depicted in Figure 1 (flow diagram). 

Table 1. In the footnote, it is better to report mean (SD) instead of mean ± SD. 

Figure 3. In the pie chart for “pCMV (≤ 90-day-old)” the Others category is missing in the legend. On the other hand, you need to be consistent with the terminology used throughout the manuscript. Please, change the heading “91 to 365-day-old” to “>90-days old”. 

Finally, it would have been interesting to have information on some maternal characteristics, like age, parity and educational level among others. Why did the authors not consider them in extracting and selecting data from the included cases?

Reviewer 2 Report

Comments and Suggestions for Authors

This is a retrospective case series of symptomatic infants with positive CMV results (viral culture, CMV-DNA PCR or CMV IgM) at some time < 1 year of age obtained in a  data set from several units in Taiwan.  Infants were classified as congenitally or postnatally infected based in the age of presentation of symptoms. They had to present with at least one of the findings defined by the authors which are classical findings of congenital CMV. 

There are major concerns with their study design that hinder comprehension of their data:

1- Why were these infants tested for CMV? Was there any criteria to test? why the testing varied? Was it related to timing at testing? Overall, how many infants in this data base had been CMV tested? Were there any infant that had other signs and symptoms which did not characterize the ones the authors listed as inclusion criteria that had been tested? This information would give the reader and idea of the relevance/magnitude of the presence of CMV-like signs and CMV infection among the infants assisted in these units, what would better support, although indirectly, their conclusions that CMV screening should be implemented in this population.

2- CMV-IgM has a low sensitivity to detect CMV infection so, those symptomatic CMV-IgM negative infants could have been erroneously classified as CMV-uninfected biasing their data. 

3- It is not clear if babies who tested positive after 3 weeks of life had tested negative within 3 weeks to exclude cCMV. if not, these infants classified as postnatally infected can be congenitally infected. So, the symptoms could have been later detected and missclassified as post natal infection. This is a very relevant point.

4-There are missing definition criteria to be clarified: a) Why intrauterine growth restriction is different form small for gestational age in Table 1?  What about perinatal CMV exposure? Was it during birth? How this was diagnosed? How sensorineural hearing loss was defined? What the neurological abnormality present? How they were defined?

5- considering the postnatal CMV infection is usually asymptomatic but the baby usually shed virus for long periods. The clinical presentation such as myocarditis, nephrotic symptom, petechia purpura, neurological abnormaties  found in > 90 days old infants could not be related to CMV-infection as shown by the authors. Have other causes ruled out?

In summary, the picture presented by the authors must be better detailed to give us  more comprehensive data. With so many missing information, the data is not useful.

Reviewer 3 Report

Comments and Suggestions for Authors

The present study retrospectively investigated the cytomegalovirus infection in infants. The study includes 272 CMV infected cases and describe their clinical presentation and follow-up. The study lack novelty but addresses the importance of CMV screening in infants. Authors need to do some modifications:

1.              Authors should provide demographic information of these infants or socioeconomic of their parents. 

2.              The authors should describe the abbreviations if used for first time such as pCMV and cCMV in abstract.

3.              Page 2 line 51 there is a typo error in reference.

4.              Authors have used t-test for analysis. Authors should check normality of the data, if data is parametric or non-parametric, as clinical data may not follow normal distribution.

5.              Authors should include and describe figure legends.

6.              In Table 1, authors could replace male sex to male.

7.              Figure 3 is a duplication of Table 2 results. Authors could remove figure 3.

Round 2

Reviewer 2 Report

Comments and Suggestions for Authors

N/A

Comments on the Quality of English Language

N/A

Reviewer 3 Report

Comments and Suggestions for Authors

No further comments

Author Response

We wanted to express our sincere gratitude for the advice you offered earlier.